# Aluminium Recycling in Single- and Multiple-Capillary Laboratory Electrolysis Cells

**Andrey Yasinskiy** [1,2]**, Sai Krishna Padamata** [1,*]**, Ilya Moiseenko** [1]**, Srecko Stopic** [2]**, Dominic Feldhaus** [2]**, Bernd Friedrich** [2] **and Peter Polyakov** [1]

[1] Laboratory of Physics and Chemistry of Metallurgical Processes and Materials, Siberian Federal University, Krasnoiarskii rabochii 95, 600025 Krasnoyarsk, Russia; ayasinskiykrsk@gmail.com (A.Y.); ilya9.97@mail.ru (I.M.); P.v.polyakov@mail.ru (P.P.)

[2] IME Process Metallurgy and Metal Recycling, RWTH Aachen University, Intzestraße 3, 52056 Aachen, Germany; sstopic@ime-aachen.de (S.S.); DFeldhaus@metallurgie.rwth-aachen.de (D.F.); bfriedrich@ime-aachen.de (B.F.)

\* Correspondence: saikrishnapadamata17@gmail.com

**Abstract:** This work is a contribution to the approach for Al purification and extraction from scrap using the thin-layer multiple-capillary molten salt electrochemical system. The single- and multiple-capillary cells were designed and used to study the kinetics of aluminium reduction in LiF–AlF$_3$ and equimolar NaCl–KCl with 10 wt.% AlF$_3$ addition at 720–850 °C. The cathodic process on the vertical liquid aluminium electrode in NaCl–KCl (+10 wt.% AlF$_3$) in the 2.5 mm length capillary had mixed kinetics with signs of both diffusion and chemical reaction control. The apparent mass transport coefficient changed from $5.6 \cdot 10^{-3}$ cm.s$^{-1}$ to $13.1 \cdot 10^{-3}$ cm.s$^{-1}$ in the mentioned temperature range. The dependence between the mass transport coefficient and temperature follows an Arrhenius-type behaviour with an activation energy equal to 60.5 kJ.mol$^{-1}$. In the multiple-capillary laboratory electrolysis cell, galvanostatic electrolysis in a 64LiF–36AlF$_3$ melt showed that the electrochemical refinery can be performed at a current density of 1 A.cm$^{-2}$ or higher with a total voltage drop of around 2.0 V and specific energy consumption of about 6–7 kWh.kg$^{-1}$. The resistance fluctuated between 0.9 and 1.4 Ω during the electrolysis depending on the current density. Thin-layer aluminium recycling and refinery seems to be a promising approach capable of producing high-purity aluminium with low specific energy consumption.

**Keywords:** aluminium; thin-layer electrolysis; molten salts; halides; capillary cell

## 1. Introduction

Aluminium is the second most utilised metal in the world, only outranked by steel, due to its outstanding mechanical and metallurgical properties. Aluminium and its alloys are extensively used in aerospace, electronics, household utensils, construction, etc. The demand for this metal is only growing, and sufficient supply would require a considerable amount of secondary aluminium production [1]. Through primary aluminium production, residues such as salt slag, aluminium dross, and red mud are generated. The environmentally hazardous residues that are generated can be drastically reduced if secondary aluminium productivity increases. Moreover, 93% of CO$_2$ emissions can be reduced using secondary aluminium production [2]. Processes such as remelting [3], electrolysis [4], and fractional solidification [5] are used for secondary aluminium production. The main disadvantage of aluminium recycling by remelting is that the recovered metal contains a significant number of impurities. Fractional solidification can produce highly pure metal, but it is difficult to accomplish on a large scale. Demand for high-purity aluminium is growing due to its wide range of applications including high-tech areas. It is used for anode foils for aluminium electrolytic capacitors, hard-disk substrates, sputtering targets, and wiring materials for semiconductor devices and liquid crystal display panels. In

the near future, new demands for such applications as compact self-ballasted fluorescent lamps, LED bulbs, solar power generation units, and wind-power generation units are expected [6]. The reasons for extensive use of high-purity aluminium are as follows [7]:

☐     oxide layers having high permittivity and insulation properties can be obtained through surface treatment;

☐     high-purity aluminium contains only a small number of impurity elements, precipitates, and inclusions;

☐     it exhibits high electrical and thermal conductivities.

Ultrapure aluminium can be produced using the Hoopes process, but the process demands a high energy consumption (18 kWh kg$^{-1}$) [8]. An ideal recycling process should be environmentally friendly, have a high metal yield with low impurities, and low energy consumption. A comparison made in [9] shows that the average energy requirement for the remelting process is 2.2 kWh/kg, while the theoretically minimal value is 510 kWh/kg. For primary aluminium production, the average and the minimum values are 26 and 10.2 kWh/kg, respectively.

A recent study suggests that platinum group metals (PGMs) can be electrochemically reduced along with aluminium from the spent catalysts [10]. The $\gamma$-Al$_2$O$_3$ catalyst carrier dissolves in fluoride melts and is electrochemically decomposed to reduce aluminium [11]. These aluminium-PGM alloys can be separated through electrochemical or pyrometallurgical processes to recover PGMs as well as the liquid aluminium [12]. Molten salt electrolysis in horizontally placed electrodes is considered one of the most promising processes for metal recovery [11]. The NaCl–KCl–AlF$_3$ (or Na$_3$AlF$_6$) melt with the addition of BaF$_2$ in the horizontal-electrode cell is used at 690–850 °C with an interelectrode distance of ~10 cm in the traditional three-layer refinery; the main drawback of this process is an extremely high energy consumption [11]. Chloride-based molten salts have also shown some promising results in aluminium extraction from aluminium alloys [13]. Yet, it still faces some problems [14] such as:

☐     extremely high hygroscopicity of AlCl$_3$;

☐     significant volatility of AlCl$_3$.

These two factors lead to the strong evaporation of electrolytes and hydrolysis of aluminium chloride in a gaseous phase. Another problem that comes from high hygroscopicity is the low corrosion resistance of cell compartments in the presence of water vapour in the molten salt.

This work is a continuation of previously attempted research [15,16], where the electrochemical reduction of aluminium was performed through a thin layer refinery. In [15,16], two different types of single-capillary cells were used to study electrode processes, and the results were comprehensively compared to those obtained in a traditional (non-capillary) cell. It was found that the addition of 10 wt.% of AlF$_3$ to an equimolar NaCl–KCl melt gives the best results among all tested compositions and allows operating the cell at a current density of 1.4 A.cm$^{-2}$. A significant problem raised from the study was that the resistance fluctuated from 0.7 to 2.5 Ω. Using a LiF–AlF$_3$ melt with relatively high electric conductivity was proposed as a means to reduce the resistance in the capillary [16]. The distance between the two liquid electrodes (aluminium alloy and pure aluminium) was reduced in an attempt to drastically minimize the Ohmic voltage drop and to increase the reaction rates due to overlapping diffusion layers. This can be achieved by the introduction of corrosion- and heat-resistant porous ceramic soaked with the electrolyte, which acts as a physical barrier between two aluminium electrodes. Cathodic processes in chloride [17,18] and fluoride [19,20] melts were studied to understand the electroreduction of aluminium. The results show the promising performance of these molten salts as they possess high electrical conductivity. The NaCl–KCl–AlF$_3$ and LiF–AlF$_3$ melts were chosen for this study based on their good performance in previous experiments [15,16]. The LiF–AlF$_3$ melts have a low liquidus temperature and high electrical conductivity. Chloride-based melts have also been considered as low-temperature electrolytes for both aluminium reduction and refinery

processes [21]. Alumina is generally not dissolved in chloride melts, although introducing a small proportion of $AlF_3$ into the melt can improve the solubility and dissolution rate of alumina.

The recycling of aluminium with capillary molten salt electrolysis can be a good alternative to other recycling processes, such as simple remelting, due to several factors. While remelting has a very low specific energy consumption, capillary thin-layer electrolysis is performed in an electrochemical cell (Al–Me)Al | $Al^{3+}$ | Al with EMF equal (or close) to zero, which has been shown in previous work [15], and which makes it possible to produce pure aluminium with much smaller energy consumption than that required for primary production. The EMF depends on the activity of aluminium in Al–Me alloy (where Me are common impurity elements such as Si, Fe, Cu), which were extensively studied [22,23]. The activity coefficient γ of Al in Al–Si is given by the equation [23]:

$$\gamma_{Al} = \frac{1}{X_{Al}} \exp\left(-\frac{\Delta G_T^0}{RT}\right) \frac{a_{Al_2O_3}^{\frac{13}{8}} \cdot a_{Si\cdot}^{\frac{3}{4}}}{a_{Al_6Si_{12}O_{13}\cdot}^{\frac{3}{8}}} \tag{1}$$

where $X_{Al}$ is the molar fraction of Al in the alloy, a is the activity, T is the temperature, R is the gas constant equal to 8.314 J.(mol.K)$^{-1}$, $\Delta G_T^0$ is the standard Gibbs energy change, and $\Delta G_T^0$ = 138 600–23.89 T (J/mol). In the case of equilibration of Al–Si alloys with their fluorides, the equation should be changed accordingly. Another factor is the absence of carbon in the process and, therefore, the absence of $CO_2$, CO, and CFx emissions, which take place in primary aluminium production.

In this paper, the effects of temperature, molten salt composition, and the number of capillaries (single vs. multiple) on the electrochemical behaviour of liquid Al electrodes were discussed. Electrochemical characterization of the single- and multiple-capillary systems using equimolar NaCl–KCl with 10 wt.% of $AlF_3$ and eutectic LiF–$AlF_3$ at 720–850 °C was performed. The results are intended to exhibit the complexity of the electrode process in the capillary electrolysis cells and to contribute to the development of the aluminium refinery technology in thin layers of molten halides. The previously used single-capillary cell [15] was improved in terms of simple manufacturing, and the pilot multiple-capillary laboratory cell was designed based on previous data.

## 2. Experiment

The NaCl–KCl–$AlF_3$ and LiF–$AlF_3$ melts were prepared from individual LiF, KCl, NaCl, and $AlF_3$ salts of reagent grade. Initially, anhydrous salts were dried at 400 °C for 4 h. Before the state of the experiments, the electrolytes were heated to operating temperature, then purified using the graphite electrode at 0.2V (vs. the $Al^{3+}$/Al potential) for 2 h. The undesirable residues were electrodeposited on the graphite surface. Two- and three-electrode cells with both electrodes being liquid aluminium were used for electrochemical measurements. The single- and multiple-capillary cells were used to perform aluminium anodic dissolution and cathodic reduction. Copper was not added to the anode, as is done in the Hoopes process, as there was no need to increase the density of the anode. The single-capillary cell consisted of a boron nitride (BN) two-electrode (working electrode/WE and reference electrode/RE) set-up placed into the graphite crucible (65 mm inner diameter, 70 mm height) filled with aluminium (counter electrode/CE) and molten NaCl–KCl–$AlF_3$ electrolyte, as shown in Figure 1.

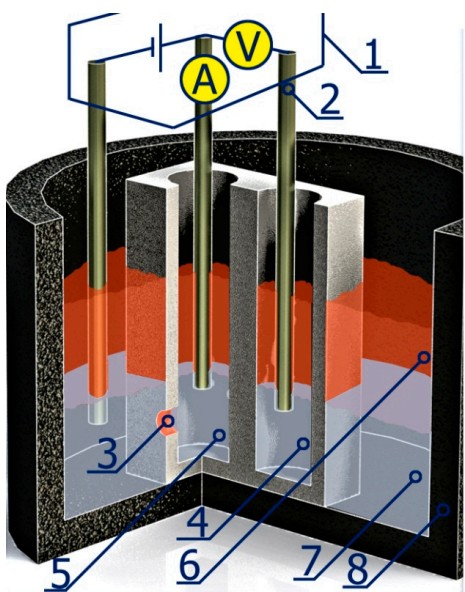

**Figure 1.** Schematic representation of a single capillary cell: 1—PGSTAT Autolab 302n, 2—tungsten current leads, 3—capillary (0.6 mm diameter, 2.5 mm length) drilled in BN two-electrode set-up, 4—Al/AlF$_3$ reference electrode, 5—Al working electrode, 6—NaCl–KCl (1:1 molar ratio) + 10 wt.%AlF$_3$, 7—Al counter electrode, 8—graphite crucible.

The BN two-electrode setup was a block (length × width × height: 30 × 15 × 72 mm$^3$) with two closed-end channels (10 mm diameter, 60 mm depth) filled with aluminium and molten salt. The tungsten rods (2 mm diameter) were immersed into aluminium in both channels to serve as current leads. The capillary was drilled out through the wall of the channel, which acted as a working electrode. The other channel was liquid-tight. However, due to the porosity of the BN block, molten salt soaked the walls and made it possible to establish an electrolytic contact between RE and WE. Due to the difference in the surface tension and wettability of the materials by aluminium and fused salt, the capillary contained only salt, with aluminium being kept outside. The apparent surface area of the electrodes in the capillary was 0.003 cm$^2$. The length of the capillary was 0.25 cm with a diameter of 0.6 mm.

In the multiple-capillary cell, a BN crucible (30 mm inner diameter and 50 mm height) with a perforated wall (10 × 20 mm$^2$) was immersed into a graphite crucible (65 mm inner diameter, 70 mm height) as shown in Figure 2. The number of capillaries was 840. Each one had a diameter of 0.04 cm and a length of 0.25 cm. The total apparent electrode surface area was 1.055 cm$^2$. Both crucibles were filled with molten aluminium and electrolytes. Two tungsten 2 mm rods were immersed into the liquid aluminium in both crucibles and acted as current leads.

The PGSTAT302n potentiostat (MetrOhm Autolab B.V., Utrecht, the Netherlands) with the 20 A booster and Nova 2.1.2 software (MetrOhm Autolab B.V., Utrecht, the Netherlands) was used to implement the studies under galvanostatic conditions. The temperature of the furnace was maintained constant by using a USB-TC01 thermocouple module (National Instruments, Austin, TX, USA) and measured using a k-type thermocouple (not shown in figures). Stationary polarization curves were obtained with current densities applied in the range of 0.07 to 15.38 A.cm$^{-2}$. The Ohmic voltage drop was determined via the I-interrupt technique. The polarization duration before the current interrupt was 120 s, and the interruption duration was 20 s. The duration between potential measurements during interruption was about 100 μs. The value of potential was taken after a rapid drop in the voltage. The experiments were conducted in an air atmosphere with temperature fluctuations of no more than ±3 °C.

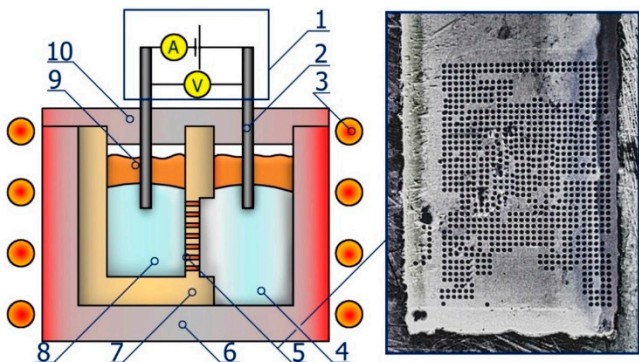

**Figure 2.** Schematic representation of a multiple-capillary cell: 1—PGSTAT Autolab 302n with BOOSTER20A, 2—tungsten current leads, 3—electric heater, 4—90Al–10Cu (wt.%) working electrode, 5—multiple-capillary wall (0.4 mm diameter, 2.5 mm length, 840 pieces), 6—outer graphite crucible, 7—inner boron nitride crucible, 8—Al counter electrode, 9—64LiF–36AlF$_3$ (molar ratio), 10—boron nitride cover.

## 3. Results and Discussion

### 3.1. Single-Capillary Electrolysis

From the previous experiments [12], it was found that an equimolar NaCl–KCl melt with 10 wt.% of AlF$_3$ had a good performance in terms of aluminium reduction kinetics. The same melt was examined to study the effects of temperature on the limiting current density. The results obtained during stationary polarization are shown in Figure 3.

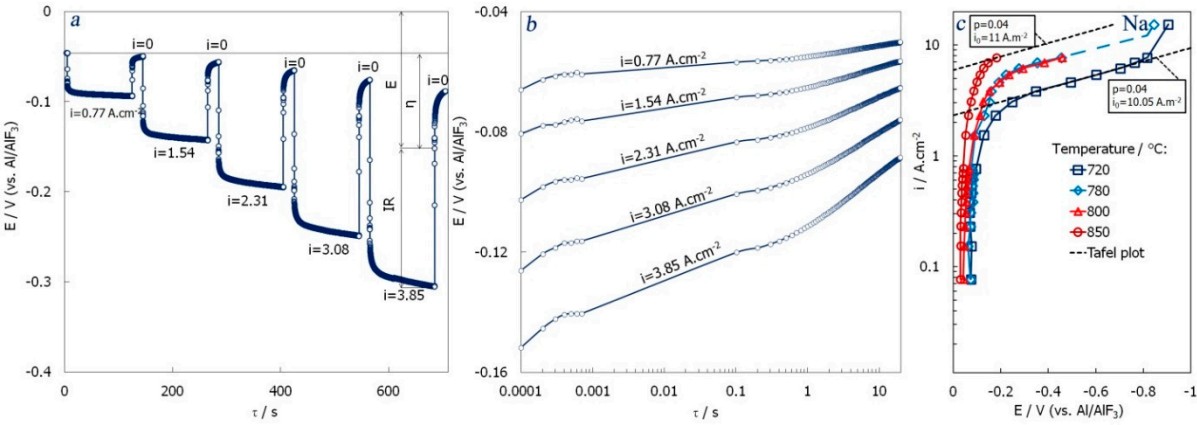

**Figure 3.** Results obtained in the single-capillary cell with the NaCl–KCl (1:1 molar ratio) + 10 wt.% AlF$_3$ melt: (**a**)—typical potential vs. time dependence obtained at 800 °C, (**b**)—potential relaxation during 20 s current interrupt, (**c**)—stationary galvanostatic polarization curves obtained at 720–850 °C.

The electrode potential was naturally shifted negative during cathodic polarization. The reversible potential was about −0.04 V vs. the Al/AlF$_3$ reference electrode. The stationary polarization curves (Figure 3c) fit the Tafel-type behaviour as they have a linear part in a wide range of current densities and potentials, which is not typical for diffusion-controlled processes. The mixed kinetics is observed due to the co-reduction of Al and alkali metals (Na or K). Co-reduction onset current density relates to the limiting current density i$_l$, which appears in the equation for diffusion-controlled processes [24]:

$$\eta_{conc} = \frac{RT}{zF} \ln\left(1 - \frac{i}{i_l}\right) \tag{2}$$

where $\eta_{conc}$ is the concentration overvoltage, z is the number of electrons transferred per one atom of Al reduced, F is the Faraday's constant equal to 96485 C.mol$^{-1}$, and i is the current density.

Since sodium is reduced along with aluminium, Na dissolves in Al and the activity of Na in Al increases. It slightly shifts the electrode potential to more negative values which can be seen in Figure 3a,b. To avoid this situation, the current density for the multiple-capillary electrolysis should be chosen below 2 A.cm$^{-2}$ at 850 °C (and less for lower temperatures), or electrolyte choice should be revised.

Another possible explanation of the linear part of the polarization curve is the activation overvoltage appearance. It is barely possible that charge transfer can be the rate-determining step at high temperatures between 720 and 850 °C. The chemical reaction control seems to be more realistic. If this is the case, then the overvoltage is governed by the equation [24]:

$$\eta_{act} = \frac{RT}{pzF}\ln i_0 + \frac{RT}{pzF}\ln i \tag{3}$$

where $\eta_{act}$ is the activation overvoltage, p is the reaction order, and $i_0$ is the exchange current density.

The theoretical curves were plotted in Figure 3c. The reaction order of 0.04 fitted the experimental data for all the temperatures, and the exchange current density increased from 10 to 11 A.m$^{-2}$ with an increase in the temperature.

The kinetic parameters obtained from the stationary polarization curves are summarized in Table 1 where $\theta$ is the temperature in °C (while T is used for the absolute temperature in K), $C_{Al^{3+}}$ is the concentration of electroactive particles, OCP is the equilibrium open circuit potential, $i_l$ is the limiting current density found by extrapolation of the linear part of the polarization curves to the OCP value, and $K_m$ is the apparent mass transport coefficient calculated according to the known relation:

$$K_m = \frac{i_l}{zFC} \tag{4}$$

**Table 1.** Kinetic parameters of aluminium reduction in the single-capillary cell.

| $\theta$, °C | $C_{Al^{3+}}$ mol.cm$^3$ | OCP, V | $i_l$, A.cm$^{-2}$ | $K_m \cdot 10^3$, cm.s$^{-1}$ |
|---|---|---|---|---|
| 720 | 0.001791 | −0.075 | 2.9 | 5.6 |
| 780 | 0.001753 | −0.073 | 4.7 | 9.3 |
| 800 | 0.001740 | −0.043 | 4.9 | 9.7 |
| 850 | 0.001709 | −0.031 | 6.5 | 13.1 |

The change in the concentration is due to a change in the molar volume of the molten salt with the temperature. The apparent mass transport coefficient vs. the temperature dependence was found to follow an Arrhenius-type relation as shown in Figure 4:

$$\left(\frac{d\ln K_m}{dT}\right) = \frac{E_A}{RT^2} \tag{5}$$

where $E_A$ is the activation energy for diffusion.

The dependence between $\ln K_m$ and $T^{-1}$ is close to linear, which allows calculating the diffusion activation energy that equals 60.5 kJ.mol$^{-1}$.

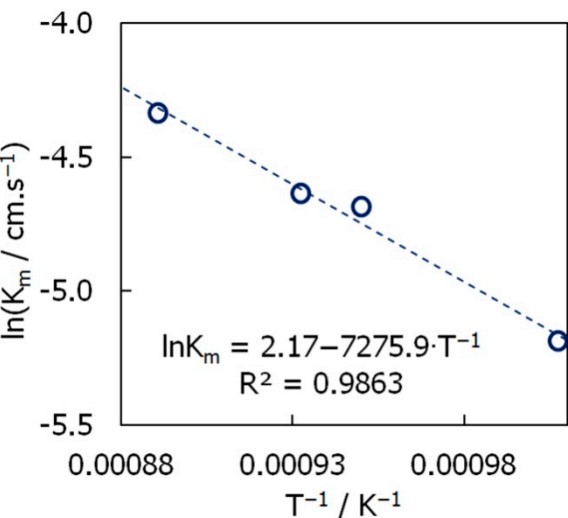

**Figure 4.** The change in the logarithm of the apparent mass transport coefficient vs. the inversed temperature.

### 3.2. Multiple-Capillary Electrolysis

The final goal was to establish the electrolysis in a multiple-capillary system to perform aluminium refinery or recycling with high productivity and low energy consumption. Keeping this in mind, the multiple-capillary laboratory cell was designed to perform small-scale short electrolysis tests. The electrolytic system was revised after the performance of the single-capillary electrolysis. Among several tested systems, which namely are NaCl–KCl–AlF$_3$, KF–AlF$_3$ [15], and LiF–AlF$_3$ [10,16], the 64LiF-34AlF$_3$ (mol. %) melt was chosen because of the huge potential window between Al and Li, low liquidus temperature, high electrical conductivity and good performance in the previous experiments [10]. The set of various current densities in the range from 0.01 to 4.74 A.cm$^{-2}$ was applied during each 180 s with current interruptions between the runs to estimate the resistance and the back EMF, which can be found from the equation:

$$\text{Back EMF} = U - IR = EMF + \eta_a + \eta_c \qquad (6)$$

where U is the total voltage, I is the current applied, R is the resistance, EMF is electromotive force (or decomposition voltage), and $\eta_a$ and $\eta_c$ are the anodic and the cathodic overvoltages, respectively. The results of the electrolysis runs are presented in Figure 5.

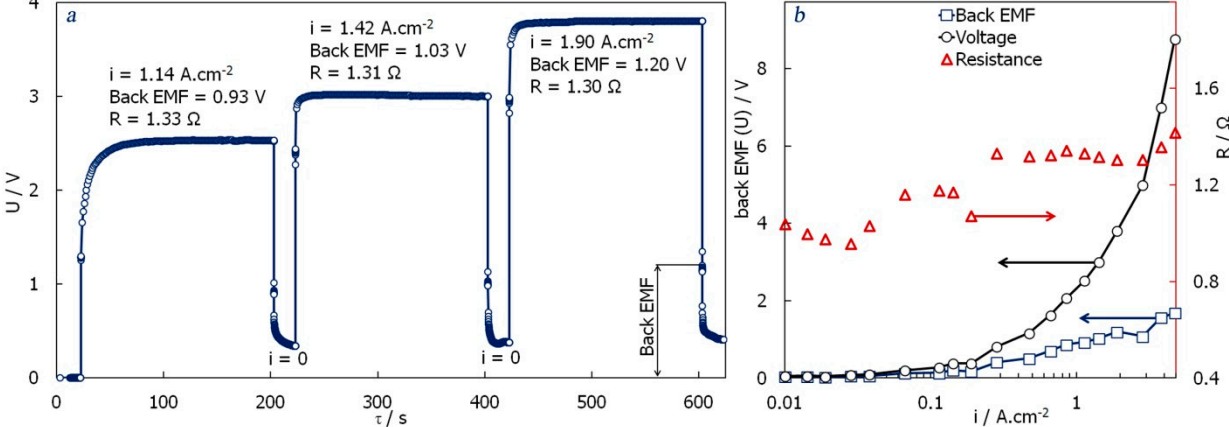

**Figure 5.** Results obtained in the multiple-capillary cell with the 64LiF–36AlF$_3$ (molar ratio) melt at 800 °C: (**a**)—typical potential vs. time dependence during galvanostatic polarization with current interrupts, (**b**)—quasi-stationary resistance, voltage, and back EMF as a function of applied current density.

The total voltage and the back EMF naturally increased with current density. The resistance changed stepwise. In one series of runs, it slightly decreased probably due to the local increase in the temperature with the current applied. There were a few leaps up in the resistance between the series of runs, which may be caused by an unexpected change in the capillary parameters due to local salt crystallization in separate capillaries during the whole experiment. From Figure 5b, at a current density of 0.9–1.1 A.cm$^{-2}$, one can expect a total voltage drop of around 2 V, which may result in a specific energy consumption of 6–7 kWh/kg at high current efficiency values above 85%. The energy requirements of this method are three times higher than those of the remelting process, which is 2.2 kWh/kg. However, the metal purity that can be obtained is much higher. While primary aluminium production requires about 26 kWh/kg, and the three-layer refinery process needs 18 kWh/kg, the capillary electrolysis process seems a promising alternative that yields high-purity aluminium with much lower specific energy consumption, which depends mainly on the specific electrical conductivity of the electrolyte and the length of capillaries, while back EMF can be rather low.

The resistance at 0.9–1.1 A.cm$^{-2}$ was about 1.3 Ω, which agrees with values obtained previously in the single-capillary cell [16]; however, it is still rather high. This value can be further reduced by optimizing the capillary parameters (the length and the diameter) and the electrolyte composition.

## 4. Conclusions

Thin-layer aluminium recycling and refinery seems to be a promising approach capable of producing high-purity aluminium with low specific energy consumption. The single-capillary, three-electrode cell was improved after previous work. The multiple-capillary laboratory electrolysis cell was first present. The main findings from the single- and multiple-capillary electrolysis are:

☐ the cathodic process on a vertical liquid-aluminium electrode in the NaCl–KCl (+10 wt.% AlF$_3$) in the 2.5 mm length capillary had mixed kinetics with signs of both diffusion and chemical reaction control;

☐ the apparent mass transport coefficient changed from $5.6 \times 10^{-3}$ cm.s$^{-1}$ to $13.1 \times 10^{-3}$ cm.s$^{-1}$, which is at least 10 times higher than usually observed in traditional molten salt cells;

☐ the dependence between the mass transport coefficient and the temperature follows an Arrhenius-type behaviour with the activation energy being 60.5 kJ.mol$^{-1}$;

☐ the presence of sodium or potassium in the electrolyte leads to the co-reduction of these metals with aluminium at relatively low current densities. For the refinery process, it is reasonable to keep the current density below 1 A.cm$^{-2}$ or consider revising the electrolyte (the LiF-AlF$_3$ was tested as a promising candidate);

☐ the galvanostatic electrolysis in the multiple-capillary cell with the 64LiF–36AlF$_3$ melt showed that the electrochemical refinery can be performed at a current density of 1 A.cm$^{-2}$, or higher, with the total voltage around 2.0 V and the specific energy consumption about 6–7 kWh.kg$^{-1}$;

☐ the resistance fluctuated between 0.9 and 1.4 Ω during the electrolysis depending on the current density.

Further efforts should be directed to the study of the effect of electrolysis conditions and capillary parameters on the extraction degree, current efficiency, and aluminium purity. The Ohmic voltage drop should also be reduced to enable refinery with a specific energy consumption of 5 kWh.kg$^{-1}$ or lower.

**Author Contributions:** Conceptualization, A.Y. and P.P.; funding acquisition, A.Y; investigation, I.M. and A.Y.; methodology, A.Y. and D.F.; supervision, P.P. and B.F.; writing—original draft, A.Y, S.K.P. and S.S. All authors have read and agreed to the published version of the manuscript.

**Funding:** The presented study was performed with the financial support of the Russian Science Foundation (Grant No. 19-79-00004).

**Institutional Review Board Statement:** Not applicable.

**Informed Consent Statement:** Not applicable.

**Data Availability Statement:** The data presented in this study are available on request from the corresponding author.

**Conflicts of Interest:** The authors declare no conflict of interest.

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
