# Peer review of "Aluminium Recycling in Single- and Multiple-Capillary Laboratory Electrolysis Cells"

_metals, doi:10.3390/met11071053_

Round 1

Reviewer 1 Report

The authors have illustrated an electrochemical method to purify aluminum from secondary waste sources. They have studied the effects of temperature, molten salt composition and number of capillaries on the overall electrochemical process. The authors can include a list of five highlights within 100 characters to define the novelty of this study. It is also advised to include a graphical abstract to pictorially summarize the study for the readers. Overall, the article has been written well with good information in the introduction section. The material and methods section needs to include details of the electrochemical cell dimensions for easy reproducibility of the process. The figure quality in the result section needs to be improved.

Author Response

Dear Reviewer,

thank you very much for your valuable comments and an invested time. We are sending our answers regarding to your questions. Attached you can find our answers to all questions by 3 Reviewers in order to understand our E-Mail.

The authors have illustrated an electrochemical method to purify aluminum from secondary waste sources. They have studied the effects of temperature, molten salt composition and number of capillaries on the overall electrochemical process. The authors can include a list of five highlights within 100 characters to define the novelty of this study.

Answer: We attached the list of highlights to this submission.

It is also advised to include a graphical abstract to pictorially summarize the study for the readers.

Answer: We agree, we added a graphical abstract to this submission.

Overall, the article has been written well with good information in the introduction section. The material and methods section needs to include details of the electrochemical cell dimensions for easy reproducibility of the process.

Answer: We agree, we added all the necessary dimensions to make possible reproducing the experiments.

The figure quality in the result section needs to be improved.

Answer: We added high quality figures with a resolution of 600 dpi as separate jpg files.

Reviewer 2 Report

Sorry but I think that the justification for the paper is weak in some places, it is possible by melting aluminium scrap to make good quaility aluminium alloys for a range of different applications. I know that while sometimes pure aluminium is used in many cases alloys are used. For example silicon is commonly added to aluminium to make the alloys used for casting the metal. By melting things which were made by casting aluminium it is possible with great ease to make ingots of aluminium alloy which are then suitable for making new objects.

I think you need to compare your energy requirements with those needed for the melting of aluminium scrap.

Now I am going to do something which might not be expected, I think that you are selling yourself short by not making the paper more clear on an important point.

Now assume that we get some aluminium ore from Jamaica and we were to convert it into aluminia (Al2O3) then if we were to convert it into aluminium in an electrochemical cell then we will consume energy and make pollution such as CF4 etc.

Now if I was to heat up aluminium car wheels and melt them then I would only need the energy needed to melt the wheels and cast new ingots.

But in your paper the electrochemical cell is 

Al | NaCl/KCl/AlF3 | Al

Thus the cell if fed with pure aluminium for both electrodes will have a emf of zero. As a result you will be able to make pure aluminium with a far smaller energy input than that required to make the metal from alumina. I think that you need to explain this in a more clear way in your paper to avoid underselling the work.

I would like to see some improvements made to the paper,

In figure 3c you only show the tafel plot for the cell going one way (going to negative vs the ref). I would like to see data where you scan both above and below the open circuit electrode potential of the working electrode. I note that the electrode potential was shifted to - 40 mV during the cathodic polarization experiment.

This way we can prove that the cell emf is zero, you could also measure the open circuit emf of the cell using a high resistance voltmeter. One interesting thing would be to measure it. You need to make a very clear case that the cell emf is a big fat zero.

You raise the question of activities of things like sodium and aluminium in the alloy. What would make me much more happy would be if you were to add an additional experiment to the paper in which you record the electrode potential of aluminium as a function of the concentration of a common alloying element in it. One common alloying element is silicon while another  is copper. If you were to do the aluminium / silicon system then you could make an estimate of the activity function for aluminum in the Al/Si alloy.

It would be interesting to know the equation for the activity function of aluminium in a Al/Si alloy. I know that for benzene in aqueous sodium chloride that the activity function is given by the following equation

Log f = k[NaCl] + k'[NaCl]^2

It would be very interesting to know what the equation would look like for aluminium in the Al / Si mixtures. This would make the paper an exceptional paper which I would enjoy reading.

I would do it at about 700 oC, I suggest you look at Bulletin of Alloy Phase Diagrams Vol. 5 No. 1 1984 

While the study is nice, I think you need to do some more work to get it ready for publication in the lab. You might also be well advised to read the literature on pyroprocessing of used nuclear fuels. 

There is the idea that used nuclear fuels can be dissolved in molten alkali metal chlorides before the uranium / plutonium / americium is electroplated onto an electrode. To my mind these molten salt alternatives to aqueous reprocessing look rather similar to what you have in mind as a means of refining aluminium. It might be a good idea to look at some of this work, I know that some work on this topic was done at the French CEA and at a EU JRC named ITU some work on americium electroplating was also done.

My report is at the border of "major changes" and reject, but if you go with reject as an editor I would regard it as a "soft reject" where the authors are strongly encouraged to do some additional experiments and then resubmit the paper.

This is in contrast to the "hard reject" which I reserve for those things which should never be published.

Author Response

Dear Reviewer,

thank you very much for your valuable comments and your invested time. We are sending our answers regarding your questions. Attached you can find all answers for 3 Reviewers in order to better understand our improvement.

Sorry but I think that the justification for the paper is weak in some places, it is possible by melting aluminium scrap to make good quaility aluminium alloys for a range of different applications. I know that while sometimes pure aluminium is used in many cases alloys are used. For example silicon is commonly added to aluminium to make the alloys used for casting the metal. By melting things which were made by casting aluminium it is possible with great ease to make ingots of aluminium alloy which are then suitable for making new objects.

Answer: We agree that justification is weak. We included more information which we believe makes it stronger. Remelting aluminium is a good approach when it comes to the production of construction parts in automotive, building or even aerospace fields. However, there is a quite huge range of areas which has a demand on high and ultra pure aluminium where remelting is not an option. These are basically high-tech applications like anode foils for aluminium electrolytic capacitors, hard-disk substrates, sputtering targets and wiring materials for semiconductor devices and liquid crystal display panels. In near future, it is expected to develop new demands for such applications as compact self-ballasted fluorescent lamps, LED bulbs, solar power generation units and wind-power generation units. High purity aluminium production is very energy-demanding now. In this article, we aim at reducing the specific energy consumption by using the new technology.

I think you need to compare your energy requirements with those needed for the melting of aluminium scrap.

Answer: We agree, we added the considerations regarding the energy requirements. The average energy requirements for the remelting process is 2.2 kWh/kg while the theoretically minimal value is 510 kWh/kg. For the primary aluminium production, the average and the minimal values are 26 and 10.2 kWh/kg. Our approach requires about 6.5 kWh/kg which is 3 times higher than remelting does, however using the first one makes sence when we need high purity.

Now I am going to do something which might not be expected, I think that you are selling yourself short by not making the paper more clear on an important point.

Now assume that we get some aluminium ore from Jamaica and we were to convert it into aluminia (Al2O3) then if we were to convert it into aluminium in an electrochemical cell then we will consume energy and make pollution such as CF4 etc.

Now if I was to heat up aluminium car wheels and melt them then I would only need the energy needed to melt the wheels and cast new ingots.

But in your paper the electrochemical cell is 

Al | NaCl/KCl/AlF3 | Al

Thus the cell if fed with pure aluminium for both electrodes will have a emf of zero. As a result you will be able to make pure aluminium with a far smaller energy input than that required to make the metal from alumina. I think that you need to explain this in a more clear way in your paper to avoid underselling the work.

Answer: Thank you for this suggestion. We completely agree with your considerations and included the explaination to the manuscript.

I would like to see some improvements made to the paper,

In figure 3c you only show the tafel plot for the cell going one way (going to negative vs the ref). I would like to see data where you scan both above and below the open circuit electrode potential of the working electrode. I note that the electrode potential was shifted to - 40 mV during the cathodic polarization experiment.

This way we can prove that the cell emf is zero, you could also measure the open circuit emf of the cell using a high resistance voltmeter. One interesting thing would be to measure it. You need to make a very clear case that the cell emf is a big fat zero.

Answer: We agree that having the EMF equal to zero is a very strong argument for using this kind of electrochemical refining. Previously, we applied both cathodic and anodic currents on the working electrode and saw that EMF was zero or close to zero in some cases. We added the discussion of it to the text with the reference to previously published work.

You raise the question of activities of things like sodium and aluminium in the alloy. What would make me much more happy would be if you were to add an additional experiment to the paper in which you record the electrode potential of aluminium as a function of the concentration of a common alloying element in it. One common alloying element is silicon while another  is copper. If you were to do the aluminium / silicon system then you could make an estimate of the activity function for aluminum in the Al/Si alloy.

Answer: We thank the Reviewer for this very nice idea. We would like to plan this research in details and perform it in near future. However we believe this quite a challenging topic to be the subject of further publication where we could discuss this question from the fundamental perspective without mixing it with the currently discussed practical investigation.

It would be interesting to know the equation for the activity function of aluminium in a Al/Si alloy. I know that for benzene in aqueous sodium chloride that the activity function is given by the following equation

Log f = k[NaCl] + k'[NaCl]^2

It would be very interesting to know what the equation would look like for aluminium in the Al / Si mixtures. This would make the paper an exceptional paper which I would enjoy reading.

I would do it at about 700 oC, I suggest you look at Bulletin of Alloy Phase Diagrams Vol. 5 No. 1 1984 

Answer: We agree that it would be quite an interesting discussion. We checked a set of publications (including the one proposed by the Reviewer) related to the Al-Si alloys and added more information related to the activities and activity coefficients of aluminium in alloys.

While the study is nice, I think you need to do some more work to get it ready for publication in the lab. You might also be well advised to read the literature on pyroprocessing of used nuclear fuels.

There is the idea that used nuclear fuels can be dissolved in molten alkali metal chlorides before the uranium / plutonium / americium is electroplated onto an electrode. To my mind these molten salt alternatives to aqueous reprocessing look rather similar to what you have in mind as a means of refining aluminium. It might be a good idea to look at some of this work, I know that some work on this topic was done at the French CEA and at a EU JRC named ITU some work on americium electroplating was also done.

Answer: We thank the Reviewer for this suggestion. We took a look at several publications on this topic. There is a huge number of work performed on electrochemistry of wasted nuclear fuel in molten salts. This is an important area nowadays. They often use LiF-based electrolyte and liquid (Bi) cathode. What different in our research is using the capillary system which we believe is very beneficial.

My report is at the border of "major changes" and reject, but if you go with reject as an editor I would regard it as a "soft reject" where the authors are strongly encouraged to do some additional experiments and then resubmit the paper.

This is in contrast to the "hard reject" which I reserve for those things which should never be published.

Reviewer 3 Report

This manuscript studied the effect of temperature, molten salt composition and the number of capillaries on the electrochemical behavior of liquid Al electrode and obtained some interesting results. I believe this manuscript could be published after considering the following points.

1) In line 55, please describe the problems faced by chloride based molten salts in detail.

2) Compared to ref12, what is the most interesting point of this manuscript? Whether the only difference between single-capillary electrolysis and multiple-capillary electrolysis is the number of capillaries, while the mechanism is exactly the same?

3) In line 66 and line 163, I think it is more suitable to add references for “ in previous experiments” and “the equation”.

4) In the Experiment section, authors heated the electrolytes to operating temperature then purified using the graphite electrode at 0.2V, but the electrolytes were prepared by reagent grade salts, hence, I want to know what are the undesirable residues. Whether it has little effect on the experiment results to remove this step?

5) In the end of section 3.1, authors obtain a diffusion activation energy that equals 60.5kj/mole, what is the theoretical signification of this value?

6) For the single-capillary electrolysis, authors adopted the equimolar NaCl-KCl melt with 10 wt.% of AlF3, while the 64LiF-34AlF3 melt was adopted for the multiple-capillary electrolysis. Apparently, therefore, it is difficult to compare the both results. Why not choose the same electrolyte melt?

Author Response

Dear Reviewer,

thank you for your  valuable comments and an invested time. We are sending our answers. Attached you can find our all answers for 3 Reviewers.

This manuscript studied the effect of temperature, molten salt composition and the number of capillaries on the electrochemical behavior of liquid Al electrode and obtained some interesting results. I believe this manuscript could be published after considering the following points.

1) In line 55, please describe the problems faced by chloride based molten salts in detail.

Answer: We added the required details to the text. Basically, there are three major problems: high volatility, high hygroscopicity of AlCl3 and low corrosion resistance of materials in the presence of dissolved water vapour in the melt.

2) Compared to ref12, what is the most interesting point of this manuscript? Whether the only difference between single-capillary electrolysis and multiple-capillary electrolysis is the number of capillaries, while the mechanism is exactly the same?

Answer: Yes, there are quite a lot of differences in these studies. The major point is that in refs. 12 and 13 we studied several different types of single-capillary cells with different set-ups of reference electrode. Different electrochemical technique (cyclic voltammetry) was applied and comparison with the results from traditional non-capillary cell) was made. In the current study, we tested the stationary potentiometry in a capillary cell and found out that this method gives more reproducible potentials due to the possibility of in-situ resistance record. Additionally, another molten salt system KF-AlF3 were tested and compared with NaCl-KCl-AlF3 with various proportions. It was found that 10wt.% of AlF3 in the chloride mixture results in higher limiting current density among other tested compositions. This was a reason of choosing this system for the current study. We added more information regarding previous studies to the text.

3) In line 66 and line 163, I think it is more suitable to add references for “ in previous experiments” and “the equation”.

Answer: We agree, we added the references.

4) In the Experiment section, authors heated the electrolytes to operating temperature then purified using the graphite electrode at 0.2V, but the electrolytes were prepared by reagent grade salts, hence, I want to know what are the undesirable residues. Whether it has little effect on the experiment results to remove this step?

Answer: These residues basically are iron, silicon and other minor metals (Ni, W, etc.) with more positive equilibrium electrode potential than that of aluminium. These metals may come to the electrolyte from reagents itself or during the preparation of the electrolyte. Usually, there is a little effect on residual currents during the measurements from these metals. However, sometimes the amount of reduced iron during the purification becomes considerable so this step is rather a protection against an unexpected situation.

5) In the end of section 3.1, authors obtain a diffusion activation energy that equals 60.5kj/mole, what is the theoretical signification of this value?

Answer: This is an interesting question. There is a possibility of calculating a theoretical value of diffusion activation energy using the first principles which we are not capable of at the moment. We also found some equations based on the correlation between kinetics and thermodynamics, however they are not very accurate and can be applied for simple systems like binary solid–solid diffusion which is not the case. Another challange is that we need to know the properties of the particle which undergoes diffusion but it is not easy because it is unknown whether the active particle is AlF4-, AlF52-, AlF63- or AlCln(3-n). In the papers on molten salt electrolysis, authors usually do not present any theoretical calculations of activation energy. However such a prediction model would be highly useful for further research. If a Reviewer has any information regarding this problem, we would be grateful if he or she shares a link to this research.

6) For the single-capillary electrolysis, authors adopted the equimolar NaCl-KCl melt with 10 wt.% of AlF3, while the 64LiF-34AlF3 melt was adopted for the multiple-capillary electrolysis. Apparently, therefore, it is difficult to compare the both results. Why not choose the same electrolyte melt?

Answer: Yes, we agree that it looks confusing. We used 64LiF – 36AlF3 in single-capillary electrolysis but it was reported in the previous publication (under ref. 13). Besides that we tested KF-AlF3 and NaCl-KCl-AlF3. We got some interesting results worth publishing in the NaCl-KCl melt but for the multiple-capillary electrolysis, we selected the option that we found the best. We modified the text to make it clearer.
